Profiling and initial validation of urinary microRNAs as biomarkers in IgA nephropathy

Wang Nannan
Bu Ru
Duan Zhiyu
Zhang Xueguang
Chen Pu
Li Zuoxiang
Wu Jie
Cai Guangyan caiguangyan@sina.com
Chen Xiangmei xmchen301@126.com
Department of Nephrology, Chinese PLA General Hospital, Chinese PLA Institute of Nephrology, State Key Laboratory of Kidney Diseases, National Clinical Research Center for Kidney Diseases , Beijing , China
Patnaik Santosh
Electronic publication date: 2015 Jun 2
Publication date: 2015
Volume: 3
Electronic Location ID: e990
Received 2015 Feb 17; Accepted 2015 May 11
Copyright: © 2015 Wang et al.
Copyright year: 2015
Copyright holder: Wang et al.
License: This is an open access article distributed under the terms of the Creative Commons Attribution License, which permits unrestricted use, distribution, reproduction and adaptation in any medium and for any purpose provided that it is properly attributed. For attribution, the original author(s), title, publication source (PeerJ) and either DOI or URL of the article must be cited.
License URL: https://creativecommons.org/licenses/by/4.0/

Keywords: MicroRNA, Microarray, Biomarkers, IgA nephropathy

Funding: 863 program 2012AA02A512 National Key Technology R&D Program 2011BAI10B00 2013BAI09B05 National Natural Science Foundation of China 81330019 81171645 973 program 2013CB530800 This study was supported by the 863 program (2012AA02A512), National Key Technology R&D Program (2011BAI10B00, 2013BAI09B05), the National Natural Science Foundation of China (81330019, 81171645) and 973 program (2013CB530800). The funders had no role in study design, data collection and analysis, decision to publish, or preparation of the manuscript.

==============================
Background. MicroRNAs (miRNAs) have been found in virtually all body fluids and used successfully as biomarkers for various diseases. Evidence indicates that miRNAs have important roles in IgA nephropathy (IgAN), a major cause of renal failure. In this study, we looked for differentially expressed miRNAs in IgAN and further evaluated the correlations between candidate miRNAs and the severity of IgAN.

Methods. Microarray and RT-qRCR (real-time quantitative polymerase chain reaction) were sequentially used to screen and further verify miRNA expression profiles in urinary sediments of IgAN patients in two independent cohorts. The screening cohort consisted of 32 urine samples from 18 patients with IgAN, 4 patients with MN (membranous nephropathy), 4 patients with MCD (minimal changes disease) and 6 healthy subjects; the validation cohort consisted of 102 IgAN patients, 41 MN patients, 27 MCD patients and 34 healthy subjects. The renal pathological lesions of patients with IgAN were evaluated according to Lee’s grading system and Oxford classification.

Results. At the screening phase, significance analysis of microarrays analysis showed that no miRNA was differentially expressed in the IgAN group compared to all control groups. But IgAN grade I–II and III subgroups (according to Lee’s grading system) shared dysregulation of two miRNAs (miR-3613-3p and miR-4668-5p). At the validation phase, RT-qPCR results showed that urinary level of miR-3613-3p was significantly lower in IgAN than that in MN, MCD and healthy controls (0.47, 0.44 and 0.24 folds, respectively, all P < 0.01 by Mann–Whitney U test); urinary level of miR-4668-5p was also significantly lower in IgAN than that in healthy controls (0.49 fold, P < 0.01). Significant correlations were found between urinary levels of miR-3613-3p with 24-hour urinary protein excretion (Spearman r = 0.50, P = 0.034), eGFR (estimated glomerular filtration rate) (r = − 0.48, P = 0.043) and Lee’s grades (r = 0.57, P = 0.014). Similarly, miR-4668-5p was significantly correlated with eGFR (r = − 0.50, P = 0.034) and Lee’s grades (r = 0.57, P = 0.013). For segmental glomerulosclerosis according to Oxford classification, patients scored as S0 had significantly lower levels of urinary miR-3613-3p and miR-4668-5p than those scored as S1 (0.41 and 0.43 folds, respectively, all P < 0.05).

Conclusions. The expression profile of miRNAs was significantly altered in urinary sediments from patients with IgAN. Urinary expression of miR-3613-3p was down-regulated in patients with IgAN. Moreover, urinary levels of both miR-3613-3p and miR-4668-5p were correlated with disease severity. Further studies are needed to explore the roles of miR-3613-3p and miR-4668-5p in the pathogenesis and progression of IgA nephropathy.

Introduction

Immunoglobulin A nephropathy (IgAN), the most common type of primary glomerulonephritis worldwide, is characterized by a predominant deposition of IgA-containing immune complexes in the glomerular mesangium (Wyatt & Julian, 2013). The diagnosis of IgAN relies entirely on a renal biopsy, which is invasive and cannot be frequently repeated in the same patient. The prognosis of this disease is good in the early stages, but over the next 20 years, up to 40% of affected patients will develop irreversible end-stage renal disease (ESRD) (Schena, 1990). Therefore, the development of non-invasive biomarkers would be of great significance for clinical assessment of IgAN.

In recent years, the role of microRNAs (miRNAs) in a variety of physiological and pathophysiological processes has received much attention. MiRNAs are a class of small non-coding RNAs that regulate gene expression at the post-transcriptional level (Esteller, 2011). It has been demonstrated that miRNAs exhibit functional dysregulation in IgAN (Chandrasekaran et al., 2012; Szeto & Li, 2014). As miRNAs are easily accessible, relatively stable and resistant to RNase-mediated degradation in body fluids (Chen et al., 2008), they have the potential to be used as non-invasive biomarkers.

Recent studies have shown that urinary levels of several selected miRNAs were significantly changed in patients with IgAN compared to healthy individuals (Wang et al., 2010; Wang et al., 2011). However, to date no data are available concerning the global urinary profile of miRNAs in IgAN patients, and a study with both healthy controls and disease controls is lacking. Therefore, the aim of this study is to find differentially expressed miRNAs in urinary sediments from IgAN patients, and further evaluate the correlations between candidate miRNAs and the severity of IgAN.

Methods

Statement on ethics

The study was carried out in accordance with the Declaration of Helsinki for Human Research and approved by the Ethics Committee of the Chinese PLA General Hospital (approval number S2014-004-02). Written informed consent for inclusion was obtained from each participant.

Study design

A screening and a validation phase were designed for the present study. Besides healthy controls, we recruited patients with membranous nephropathy (MN) or minimal change disease (MCD) as other glomerulopathy controls. In the screening phase, 32 urine samples collected from 18 patients with IgAN and 14 controls were subjected to Affymetrix GeneChip miRNA 4.0 Array to identify miRNAs that were significantly differentially expressed. In the validation phase, the expression of different miRNAs was validated by real-time quantitative polymerase chain reaction (RT-qPCR) in samples from 102 patients with IgAN, 41 patients with MN, 27 patients with MCD and 34 healthy controls.

Enrollment of subjects

Patients who received renal biopsy for the first time from December 2013 to November 2014 at the Nephrology Department of the Chinese PLA General Hospital were consecutively recruited for the present study. The diagnosis was confirmed based on clinical findings and renal biopsies. A total of 322 consecutive patients were diagnosed as IgAN, MN or MCD. Among them, we excluded 41 patients because they were <20 or >50 years old at renal biopsy, 26 patients because they had been treated with steroids within 6 months prior to renal biopsy, and 7 patients because there were less than eight glomeruli in light microscopy sections. We also excluded patients with a concomitant diagnosis of chronic hepatic disease (n = 14), diabetes (n = 11), urinary, respiratory or gastrointestinal tract infection (n = 7), rheumatoid arthritis (n = 3), Henoch-Schonlein purpura (n = 7) and systemic lupus erythematosus (n = 10).

Healthy volunteers who were between 20 and 50 years old were recruited at the Physical Examination Center of the same hospital. All of them had normal renal function, normal urinalysis results, and no personal or family history of nephropathy or other serious illness.

Finally, 120 patients with IgAN, 45 patients with MN, 31 patients with MCD and 40 healthy volunteers were enrolled in this study. Among the IgAN patients, the first 6 patients with grade I–II according to Lee’s grading system who met the inclusion criteria and a sex ratio of 1–1 were selected as a screening cohort, as well as 6 patients with grade III and 6 patients with grade IV–V. In addition, 14 age- and gender-matched controls which included 4 patients with MN, 4 patients with MCD and 6 healthy volunteers were selected in the screening cohort.

The demographic and clinical data, such as age, gender, mean aortic pressure (MAP), serum creatinine (Scr) and 24-hour urinary protein excretion (UPE), of all included participants were recorded at the time of kidney biopsy. The estimated glomerular filtration rate (eGFR) was calculated using the Chronic Kidney Disease Epidemiology Collaboration (CKD–EPI) equations (Levey et al., 2009).

Evaluation of renal pathological lesion

Renal tissues obtained by biopsy were stained with haematoxylin and eosin (H &E), periodic acid Schiff (PAS), periodic acid silver methenamine (PASM) and Masson’s trichrome for light microscopy and with IgG, IgA, IgM, C3 and C1q for immunofluorescence. Two pathologists who were blinded to patients’ data evaluated the renal biopsy slides separately.

For patients with IgAN, the severity of renal pathological lesions were evaluated according to Lee’s grading system (Lee et al., 1982) and Oxford classification (Roberts et al., 2009). First, all cases of IgAN were categorized into grade I–V according to Lee’s grading system. Then, four variables of Oxford classification were further evaluated; that is, mesangial proliferation (M1—more than half the glomeruli have more than three cells in a mesangial area, M0—not meet the criteria of M1), endocapillary hypercellularity (E1—present, E0—absent), segmental glomerulosclerosis (S1—present, S0—absent ), and tubular atrophy/interstitial fibrosis (T0—0%–25%, T1—26%–50%, T2—>50%).

Urine sample preparation

A whole-stream early morning urine specimen was collected at the day of renal biopsy. The urine sample was centrifuged at 3,000 g for 30 min and at 13,000 g for 5 min at 4 °C; supernatant was discarded (Wang et al., 2012; Wang & Szeto, 2013). Then the urinary cell pellet was lysed by RNA lysis buffer MZ (catalog number DP501; Tiangen Biotech, Beijing, China) and stored at −80 °C until used.

RNA extraction

Total RNA was extracted using a miRcute miRNA Isolation Kit (catalog number DP501; Tiangen Biotech, Beijing, China) according to the manufacturer’s protocol. The quantity (ng/ml) and purity (ratio of absorbances of the RNA isolates at 260 nm and 280 nm [A260/280]) of the RNAs obtained was evaluated by NanoDrop 2000 spectrophotometer (Thermo Scientific, Waltham, Massachusetts, USA).

MiRNA microarray analysis

Affymetrix GeneChip miRNA 4.0 Array (Affymetrix, Santa Clara, California, US, catalog number 902411), which covers all of the 2,578 mature human miRNAs available in miRBase version 20 (June 24, 2013, www.mirbase.org/) (Kozomara & Griffiths-Jones, 2011), was used to profile miRNA expressions. Briefly, 1 µg of RNA was polyA tailed and labelled with a FlashTag Biotin HSR RNA Labeling Kit (Affymetrix, Santa Clara, California, US, catalog number 901911). The labelled RNA was hybridized at 48 °C for 16 h on the miRNA arrays, which were washed and stained with Affymetrix Fluidics Station 450 (Affymetrix, Santa Clara, California, US) and scanned with an Affymetrix GeneChip Scanner 3000 (Affymetrix, Santa Clara, California, US) using the Command Console software (Affymetrix, Santa Clara, California, US). The data were analyzed with miRNA QCTool (Affymetrix, Santa Clara, California, US) using the Affymetrix default analysis settings and quantile as the normalization method. The microarray data have been deposited in Gene Expression Omnibus (GEO) at NCBI (http://www.ncbi.nlm.nih.gov/geo/) with accession number GSE64306.

RT-qPCR analysis

RT-qPCR was carried out in compliance with the MIQE guidelines (Bustin et al., 2009) to verify the candidate miRNAs revealed by microarray. Briefly, reverse transcription was performed using 500 ng of total RNA and a miRcute miRNA First-Strand cDNA Synthesis Kit (catalog number KR201; Tiangen Biotech, Beijing, China) according to the manufacturer’s protocol. RT-qPCR was performed on an Applied Biosystems 7500 Fast Dx Real-Time PCR Instrument (Applied Biosystems, Carlsbad, Calfornia, USA) using a miRcute miRNA qPCR Detection Kit (catalog number FP401; Tiangen Biotech, Beijing, China), according to the manufacturer’s instructions. All of the primers were purchased from Tiangen Biotech Company (Beijing, China). All PCR reactions were performed in triplicate, followed by melt curve analysis to verify their specificity and identity. U6 was selected as the endogenous reference control (Mestdagh et al., 2009). Relative miRNA expression levels were calculated using the ΔΔCt method as previously described (Livak & Schmittgen, 2001).

MiRNA target prediction and function analysis

The candidate miRNAs were imported into the miRWalk algorithm (Dweep et al., 2011) (http://www.umm.uni-heidelberg.de/apps/zmf/mirwalk/), and the prediction of their target genes was performed using nine additional algorithms, including TargetScan, DIANAmT, miRanda, miRDB, RNAhybrid, PICTAR4, PICTAR5, PITA and RNA22.

To explore the functional annotation and pathway enrichment of those predicted genes, the Gene Ontology (GO) and KEGG (Kyoto Encyclopedia of Genes and Genomes) database analyses were conducted using a DAVID (the Database for Annotation, Visualization and Integrated Discovery) online analysis tool (Huang da, Sherman & Lempicki, 2009) (http://david.abcc.ncifcrf.gov/tools.jsp), in which we focused on the GO biology process (BP) feature.

Statistical analysis

The microarray data were analyzed by the algorithm of SAM (significance analysis of microarrays, www-stat.stanford.edu/~tibs/SAM/) (Olson, 2006). The false discovery rate (FDR) was set to <0.05 and the minimum fold change (FC) was set to >2.0 or <0.5. Hierarchical clustering was carried out using the MeV 4.9 software (Multi Experiment Viewer, http://www.tm4.org/mev.htm) to generate both miRNA and sample trees based on Pearson correlation. For the RT-qPCR data, statistical analysis was performed by SPSS version 20.0 (SPSS, USA). Data were compared by Mann–Whitney U test, Kruskal–Wallis test and Spearman’s rank order correlations as appropriate. In bioinformatics analysis, the Fisher’s exact test and χ2 test were used to select the significant GO category or KEGG pathway, and the FDR was calculated to correct the P value. A P-value <0.05 was considered to be statistically significant.

Results

Patients characteristics

Demographic and clinical characteristics of the screening and validation cohorts at the time of renal biopsy are provided in Tables 1 and 2 respectively. There were no significant differences in age and MAP among different groups in both cohorts.

Table 1 Demographic and clinical data of subjects in the screening cohort.

Continuous data are presented as mean ± SEM; categorical data are presented as counts.

	IgAN I–II	IgAN III	IgAN IV–V	MN	MCD	HC	
Cases	6	6	6	4	4	6	
Gender (M/F)	3/3	3/3	3/3	2/2	2/2	3/3	
Age (years)	27 ± 5.3	32.3 ± 9.3	36.3 ± 10.5	25.8 ± 3.1	33.3 ± 8.5	31.2 ± 8.4	
MAP (mmHg)	88.78 ± 5.01	90.89 ± 2.93	101 ± 9.3	91.42 ± 10.85	91.92 ± 14.71	87.21 ± 4.87	
UPE (g/day)	1.21 ± 1.78	0.82 ± 0.75	1.73 ± 0.52	4.17 ± 1.66	8.15 ± 2.22	0.008 ± 0.003	
Scr (μmol/L)	78.7 ± 20.22	87.3 ± 35.56	126.23 ± 23.12	61.63 ± 19.50	82.58 ± 25.63	59.14 ± 10.25	
eGFR (ml/min/1.73 m2)	102.93 ± 15.89	94.64 ± 23.97	57.57 ± 17.48	129.29 ± 17.54	97.48 ± 28.47	135.46 ± 12.87	
Oxford classification							
M0/M1	6/0	2/4	3/3	–	–	–	
E0/E1	6/0	5/1	6/0	–	–	–	
S0/S1	4/2	1/5	0/6	–	–	–	
T0/T1/T2	6/0/0	3/2/1	0/1/5	–	–	–	
Notes.

IgAN IgA nephropathy

MN membranous nephropathy

MCD minimal change disease

HC healthy control

MAP mean aortic pressure

UPE 24-hour urinary protein excretion

Scr serum creatinine

eGFR estimated glomerular filtration rate

Table 2 Demographic and clinical data of subjects in the validation cohort.

Continuous data are presented as mean ± SEM; categorical data are presented as counts and/or proportions.

	IgAN	MN	MCD	HC	P value	
					IgAN vs. MN	IgAN vs. MCD	IgAN vs. HC	
Cases	102	41	27	34	–	–	–	
Gender (M/F)	57(55.77%)	24(58.54%)	69(22.22%)	17(50%)	0.857	0.008	0.489	
Age (years)	35.87 ± 11.82	35.8 ± 9.3	31.6 ± 10.9	33.1 ± 7.9	0.935	0.963	0.659	
MAP (mmHg)	95.53 ± 12.16	94.17 ± 9.72	88.69 ± 10.89	86.59 ± 7.35	0.763	0.149	0.061	
UPE (g/day)	1.57 ± 1.28	4.61 ± 2.61	7.01 ± 2.86	0.015 ± 0.013	<0.001	<0.001	<0.001	
Scr (μmol/L)	105.89 ± 51.68	69.47 ± 17.05	78.08 ± 22.12	61.0 ± 11.89	<0.001	0.003	<0.001	
eGFR (ml/min/1.73 m2)	80.76 ± 29.65	119.62 ± 18.77	100.87 ± 24.56	126.54 ± 10.85	<0.001	0.038	<0.001	
Notes.

IgAN IgA nephropathy

MN membranous nephropathy

MCD minimal change disease

HC healthy control

MAP mean aortic pressure

UPE 24-hour urinary protein excretion

Scr serum creatinine

eGFR estimated glomerular filtration rate

In the screening cohort, the levels of UPE in IgAN grade I–II and III subgroups were lower than that in MCD group (all P < 0.01) and higher than that in healthy control (HC) group (all P < 0.01); the levels of Scr in IgAN grade IV–V subgroup were higher than that in both MN and HC group (all P < 0.05), whereas the levels of eGFR showed an opposite trend.

In the validation cohort, the levels of UPE in IgAN group were lower than that in MN and MCD group (all P < 0.01), but higher than that in HC group (P < 0.001); the levels of Scr were higher than that in all control groups (all P < 0.01), whereas the levels of eGFR showed an opposite trend. The histopathological parameters are shown in Table 3.

Table 3 Histopathological characteristics of patients with IgAN.

According to the Oxford classification, four variables were evaluated; that is, mesangial proliferation (M), segmental glomerulosclerosis (S), endocapillary hypercellularity (E) and tubular atrophy/interstitial fibrosis (T).

Oxford classification	Lee’s grade	Total	
	I	II	III	IV	V		
M0/M1	4/0	17/0	31/21	6/18	1/4	59/43	
E0/E1	4/0	15/2	43/9	22/2	5/0	89/13	
S0/S1	4/0	14/3	12/40	3/21	0/5	33/69	
T0/T1/T2	4/0/0	14/3/0	38/14/0	0/20/4	0/0/5	56/37/9	
Total	4	17	52	24	5	102	
Notes.

IgAN IgA nephropathy

The yield of RNA from urinary sediments

RNA yield from urinary sediments (ng of RNA isolated from per ml of urine) was determined and showed no significant differences among patients with IgAN, MN, MCD and healthy subjects. The overall median was 22.73 ng/ml (interquartile range [IQR] = 10.75–40.35). A260/280 was also recorded and showed no significant differences among different groups in the present study. The overall mean of A260/280 was 1.95 (range = 1.82–2.07, SD = 0.06).

Differentially expressed miRNAs in the screening phase

Global miRNA profiling from microarray analysis

To identify miRNAs differentially expressed in IgAN, miRNA microarrays were used to analyze their global expression profile of the four groups (IgAN, MN, MCD and HC). The 32 urine samples of the screening cohort were profiled on Affymetrix GeneChip miRNA 4.0 Arrays.

Among 2,578 mature human miRNAs represented on the microarray, 780 (30.3%) miRNAs were identified as expressed in at least one urine sample. Unsupervised hierarchical clustering analysis demonstrated that miRNA profiling clearly differentiated IgAN samples from healthy controls; there were, however, one MN sample and two MCD samples mixed with IgAN, indicating that the differences between IgAN patients and disease controls were not as clear (Fig. 1).

Figure 1 Global urinary miRNA profiling in patients with IgAN and controls.

Unsupervised hierarchical clustering using log2-transformed microarray signal values of the 780 expressed miRNAs is shown as a heat-map. Columns represent urine samples and rows miRNAs. Urine samples were from 18 patients with IgAN (red), 6 healthy subjects (orange), 4 patients with MN (dark blue) and 4 patients with MCD (light blue). Heat-map colours represent relative miRNA expression as indicated in the colour scale: yellow represents high expression; blue represents low expression; black respresents median expression level equal to 1. HC, healthy control; IgAN, IgA nephropathy; I–II, III and IV–V, grade I–II, III and IV–V according to Lee’s grading system, respectively; MN, membranous nephropathy; MCD, minimal change disease.

Differentially expressed miRNAs in IgAN

The microarray data were analyzed by SAM; the results identified various differentially expressed miRNAs that presented a FC >2 or <0.5 and P value <0.05 in urinary sediments of IgAN samples. Among them, there were 117 miRNAs differentially expressed between IgAN and HC (Table S1); 78 miRNAs differentially expressed between IgAN and MN (Table S2); and 11 miRNAs differentially expressed between IgAN and MCD (Table S3). There was not a single miRNA, however, that showed significantly different expression across all four groups.

Differentially expressed miRNAs in IgAN subgroups

In the screening phase, we grouped IgAN grade I–II as the early pathological change group; grade III as the mild pathological change group; and grade IV–V as the severe group according to Lee’s grading system. Then we compared the different subgroups of IgAN (grade I–II, III and IV–V) with both healthy and disease controls. The differentially expressed miRNAs are shown in the form of volcano plot (Fig. S1) and Venn diagram (Fig. 2). All these miRNAs presented a FC >2 or <0.5 and P value <0.05.

Figure 2 Overlapping relationship of the differentially expressed miRNAs in IgAN subgroups.

Urinary levels of miRNAs in the subgroups of IgAN (IgAN grade I–II, III and IV–V subgroup according to Lee’s grading system) were compared with each control group (HC, MN and MCD) in the screening phase. All differentially expressed miRNAs presented a fold change >2 and P value <0.05. The miRNAs in the centre of Venn diagram were (A) miR-223-3p, miR-629-5p, miR-3613-3p and miR-4668-5p; (B) miR-150-5p, miR-572, miR-371b-5p, miR-3613-3p, miR-4668-5p and miR-6750-5p. IgAN, IgA nephropathy; HC, healthy control; MN, membranous nephropathy; MCD, minimal change disease.

In the IgAN grade I–II subgroup, two miRNAs (hsa-miR-223-3p, hsa-miR-629-5p) were in high levels and two (hsa-miR-3613-3p, hsa-miR-4668-5p) were in low levels compared to all of the control groups (Fig. 2A). In the IgAN grade III subgroup, three miRNAs (hsa-miR-150-5p, hsa-miR-572 and hsa-miR-371b-5p) were in high levels and three (hsa-miR-3613-3p, hsa-miR-4668-5p and hsa-miR-6750-5p) were in low levels compared to all control groups (Fig. 2B). IgAN grade I–II and III subgroups shared dysregulation of only two miRNAs (miR-3613-3p and miR-4668-5p). Interestingly, there were no overlaps in changes to miRNAs for IgAN grade IV–V subgroup as shown in the Venn diagram (Fig. 2C). It turned out that miRNAs were in different levels only in specific IgAN subgroups.

Validation of differentially expressed miRNAs using RT-qPCR

To check the accuracy of the microarray-based miRNA quantification, the eight above-mentioned miRNAs (miR-223-3p, miR-629-5p, miR-150-5p, miR-572, miR-371b-5p, miR-3613-3p, miR-4668-5p and miR-6750-5p) were re-examined using RT-qPCR in the 32 samples of the screening cohort. We then analyzed correlations between log2-transformed microarray signal and RT-qPCR Ct values.

As shown in Table 4, seven of the eight miRNAs (miR-223-3p, miR-629-5p, miR-150-5p, miR-572, miR-371b-5p, miR-3613-3p and miR-4668-5p) demonstrated significant correlations (|r| > 0.7, P < 0.01) between the two sets of measurements, indicating that the microarray data was relatively reliable.

Table 4 Correlations between miRNA quantification by microarray and RT-qPCR.

miRNA	r	95% CI	P value	
miR-223-3p	−0.86	−0.93∼−0.72	<0.001	
miR-629-5p	−0.74	−0.87∼−0.52	<0.001	
miR-150-5p	−0.93	−0.97∼−0.8	<0.001	
miR-572	−0.82	−0.91∼−0.68	<0.001	
miR-371b-5p	−0.77	−0.88∼−0.58	<0.001	
miR-3613-3p	−0.84	−0.94∼−0.66	<0.001	
miR-4668-5p	−0.72	−0.88∼−0.46	<0.001	
miR-6750-5p	−0.37	−0.67∼−0.01	0.04	
Notes.

RT-qPCR real-time quantitative polymerase chain reaction

r Spearman correlation coefficient

95% CI 95% confidence interval

Analysis of candidate miRNAs in the validation phase

Validation of candidate miRNAs using RT-qRCR

For confirmation purposes, the two miRNAs (miR-3613-3p and miR-4668-5p) that were differentially expressed in both IgAN grade I–II and III subgroups in the screening phase were selected as candidates and analyzed in the validation cohort, which included 102 patients with IgAN, 41 patients with MN, 27 patients with MCD and 34 healthy controls.

RT-qPCR results showed that the levels of miR-3613-3p were significantly lower in IgAN as compared with that in HC, MN and MCD, respectively [0.27 (0.19–0.38) vs. 1.10 (0.9–1.26), 0.58 (0.32–1.00) and 0.61 (0.39–0.77), respectively, all P < 0.01] (Fig. 3A); the levels of miR-4668-5p were significantly lower in IgAN than that in HC [0.54 (0.38–0.60) vs. 1.11 (0.73–1.22), P = 0.001]; there were no significant differences for miR-4668-5p, however, between IgAN and the disease controls (Fig. 3B).

Figure 3 Comparison of candidate miRNAs levels between IgAN and each control group.

Urinary levels of candidate miRNAs were analyzed by RT-qPCR in the validation cohort, which including 102 patients with IgAN, 41 patients with MN, 27 patients with MCD and 34 healthy controls. The whisker-box plots depict the relative expression level of miR-3613-3p and miR-4668-5p. The boxes indicate median and 25th and 75th percentiles; whisker caps indicate 5th and 95th percentiles. Statistically significant differences were determined by Mann–Whitney U test. Levels are represented as ratio to the median of healthy controls. NS, not significant (P > 0.05); IgAN, IgA nephropathy; HC, healthy control; MN, membranous nephropathy; MCD, minimal change disease. (A) Urinary miR-3613-3p levels was significantly lower in IgAN as compared with that in HC, MN and MCD, respectively [0.27 (0.19–0.38), n = 102 vs. 1.10 (0.9–1.26), n = 34; 0.58 (0.32–1.00), n = 41 and 0.61 (0.39–0.77), n = 27; respectively, all P < 0.01]. (B) Urinary miR-4668-5p levels were significantly lower in IgAN than that in HC [0.54 (0.38–0.60), n = 102 vs. 1.11 (0.73–1.22), n = 34; P = 0.001]. Urinary miR-4668-5p levels there were were not different between IgAN and MN and MCD [0.54 (0.38–0.60), n = 102 vs. 0.56 (0.41–0.88), n = 41 and 0.65 (0.49–0.87), n = 27; repectively].

Correlations between candidate miRNAs and clinical parameters

Correlations between urinary levels of candidate miRNAs and clinical parameters, such as UPE, Scr, eGFR and Lee’s grades, were further analyzed within the IgAN group (Table 5). Urinary levels of miR-3613-3p were positively correlated with both UPE (r = 0.50, P = 0.034) and Lee’s grades (r = 0.57, P = 0.014), while negatively correlated with eGFR (r = − 0.48, P = 0.043). Similarly, significant correlations were found between urinary levels of miR-4668-5p with eGFR (r = − 0.50, P = 0.034) and Lee’s grades (r = 0.57, P = 0.013).

Table 5 Correlations between candidate miRNAs and clinical parameters in patients with IgA nephropathy.

	miR-3613-3p	miR-4668-5p	
	r	95% CI	P value	r	95% CI	P value	
Age	0.12	−0.37∼0.54	0.631	0.26	−0.28∼0.67	0.29	
MAP	0.42	−0.06∼0.72	0.084	0.34	−0.19∼0.73	0.173	
UPE	0.501	0.03∼0.83	0.034	0.44	−0.05∼0.76	0.066	
Scr	0.451	−0.06∼0.79	0.06	0.42	−0.06∼0.73	0.083	
eGFR	−0.48	−0.84∼0.02	0.043	−0.5	−0.84∼0.08	0.034	
Lee’s grades	0.57	0.07∼0.84	0.014	0.57	0.04∼0.85	0.013	
Notes.

r Spearman correlation coefficient

95% CI 95% confidence interval

MAP mean aortic pressure

UPE 24-hour urinary protein excretion

Scr serum creatinine

eGFR estimated glomerular filtration rate

In addition, relationships between expressions of candidate miRNAs and four components of Oxford classification are shown in Fig. 4. For segmental glomerulosclerosis, patients scored as S0 had significantly lower levels of urinary miR-3613-3p and miR-4668-5p than those scored as S1 [0.23 (0.12–0.33) vs. 0.56 (0.29–2.50), 0.23 (0.18–0.25) vs. 0.53 (0.35–1.90), respectively, all P < 0.05].

Figure 4 Relationships between candidate miRNAs levels and pathological parameters in patients with IgAN according to Oxford classification.

The whisker-box plots with medians and inter-quartile ranges depict the relative expression level of miR-3613-3p and miR-4668-5p. Data were analyzed by Mann–Whitney U test. IgAN, IgA nephropathy; HC, healthy control; MN, membranous nephropathy; MCD, minimal change disease. (A) Urinary miR-3613-3p levels of patients with S1 were higher than patients with S0 [0.56 (0.29–2.5), n = 69 vs. 0.23 (0.12–0.33), n = 33; P = 0.019]. Urinary miR-3613-3p levels of patients with M1/M0, E1/E0 and T2/T1/T0 were not different between each other [0.28 (0.21–1.52), n = 43 vs. 0.51 (0.31–1.3), n = 59; 0.55 (0.26–1.61), n = 13 vs. 0.49 (0.26–1.22), n = 89; 2.58 (0.81–9.18), n = 9 vs. 0.72 (0.5–1.3) vs. 0.33 (0.23–0.51), n = 56]. (B) Urinary miR-4668-5p levels of patients with S1 were higher than patients with S0 [0.53 (0.35–1.9), n = 69 vs. 0.23 (0.18–0.25), n = 33; P = 0.019]. Urinary miR-4668-5p levels of patients with M1/M0, E1/E0 and T2/T1/T0 were not different between each other [0.33 (0.26–1.55), n = 43 vs. 0.41 (0.3–0.94), n = 59; 0.43 (0.32–1.55), n = 13 vs. 0.4 (0.24–0.94), n = 89; 2.18 (0.73–3.13), n = 9 vs. 0.43 (0.36–0.89) vs. 0.35 (0.23–0.41), n = 56].

Bioinformatics analysis of candidate miRNAs

Predicted target genes of candidate miRNAs

Only 3 out of 10 algorithms of miRWalk (DIANAmT, miRDB and Targetscan) can query the predicted target genes of miR-3613-3p and miR-4668-5p. For miR-3613-3p, there were 5,138, 2,292 and 1,249 target genes predicted by DIANAmT, miRDB and Targetscan, respectively. For miR-4668-5p, there were 1,766, 1,169 and 552 target genes predicted by DIANAmT, miRDB and Targetscan, respectively. To reduce redundancy, only genes predicted by all three algorithms were selected for further analysis; that is, 593 predicted targets genes for miR-3613-3p and 175 predicted targets genes for miR-4668-5p (data not shown).

Functions and pathway analysis of miRNAs targets

To explore in-depth biological information, the GO and KEGG database analyses were conducted using a DAVID online analysis tool. All of the significant GO Biological Processes for miR-3613-3p and miR-4668-5p are listed in Table S4 and Table S5 respectively; the top 20 GO Biological Processes according to P values and all significant KEGG pathways for miR-3613-3p and miR-4668-5p are shown in Fig. 5 and Fig. 6 respectively.

Figure 5 GO biology process and KEGG pathway enrichment analysis for predicted miRNA targets of miR-3613-3p.

GO biology process and KEGG pathway enrichments were performed by mapping the predicted target genes using DAVID online analysis tool. P < 0.05 was used as a threshold to select significant GO terms and KEGG pathways. −lg(P value) is the negative log10 of the P value. The top 20 GO biology processes accordign to −lg(P value) and a total of 18 KEGG pathways for miR-3613-3p were annotated. GO, Gene Ontology; BP, biology process; KEGG, Kyoto Encyclopedia of Genes and Genomes.

Figure 6 GO biology process and KEGG pathway enrichment analysis for predicted miRNA targets of miR-4668-5p.

GO biology process and KEGG pathway enrichments were performed by mapping the predicted target genes using DAVID online analysis tool. P < 0.05 was used as a threshold to select significant GO terms and KEGG pathways. −lg(P value) is the negative log10 of the P value. The top 20 GO biology processes accordign to −lg(P value) and a total of 2 KEGG pathways for miR-4668-5p were annotated. GO, Gene Ontology; BP, biology process; KEGG, Kyoto Encyclopedia of Genes and Genomes.

There were 254 and 49 significant GO terms for miR-3613-3p and miR-4668-5p respectively (P < 0.05). The most significantly enriched GO terms for miR-3613-3p were the regulation of transcription (GO: 0045449, GO: 0006350) and macromolecule metabolic process (GO: 0010604, GO: 0010557) as shown in Fig. 5A; it also had significant enrichment in B cell differentiation (GO: 0030183, P = 0.006) and activation (GO: 0042113, P = 0.016) as listed in Table S4. Similarly, the most significantly enriched GO terms for miR-4668-5p (as shown in Fig. 6A) also included transcription (GO: 0006350, GO: 0045449) and macromolecule metabolic process (GO: 0051252, GO: 0010558), as well as nuclear transport (GO: 0006607, GO: 0051169).

In addition, as shown in Figs. 5B and 6B, there were 18 and 2 significant KEGG pathways for miR-3613-3p and miR-4668-5p respectively (P < 0.05); both included Wnt signaling pathway (hsa04310, P = 0.012 and P = 0.013 respectively).

Discussion

Genome-wide analyses of miRNA expressions in peripheral blood mononuclear cells (PBMCs) and kidney biopsy tissues have identified a number of miRNAs differentially expressed in patients with IgAN compared to healthy controls (Serino et al., 2012; Tan et al., 2013). In the present study, our microarray data also showed that there were a number of miRNAs that were up-regulated in urinary sediments of IgAN patients when compared with healthy subjects. However, the differentially expressed miRNAs identified in the three studies were not completely consistent with each other, which may be due to the use of different tissues and/or the possible effect of the different races of the subjects. It is therefore important to investigate miRNA profiles from different races in a larger study.

Wang et al. (2011) and Wang et al. (2012) quantified urinary levels of several miRNAs in 43 patients with IgAN and 13 healthy volunteers and found that the levels of miR-146a, miR-155 and miR-93 of IgAN were significantly higher than those of healthy controls, which are consistent with our microarray results. However, these studies lacked disease controls to prove whether the differentially expressed miRNAs are disease-specific for IgAN. In theory, any aberrant miRNA expression observed in IgAN patients could be either disease-specific or present in all patients with chronic kidney disease (Szeto & Li, 2014). Thus, in the present study, we recruited patients with MN or MCD as other glomerulopathy controls and explored the whole urinary miRNA profile in IgAN patients.

Analyses of our microarray data demonstrated that although the urinary miRNA profilings of IgAN samples were clearly differentiated from healthy controls, they were mixed to a certain degree with disease controls; that is, no miRNA was found to be disease-specific for IgAN at the screening phase of this study. Thus, we performed subgroup analyses according to Lee’s grading system, the results showed that eight miRNAs were differentially expressed either in the IgAN grade I–II subgroup or in the IgAN grade III subgroup compared to all control groups, but no miRNA was specific in the IgAN grade IV–V subgroup. This result may be explained by considerable histologic variability of IgAN, which ranges from no detectable histologic lesion to diffuse proliferative glomerulonephritis, and these cumulative changes may eventually results in glomerulosclerosis and tubulointerstitial fibrosis that are common in patients with chronic kidney disease (Haas & Reich, 2012; Loeffler & Wolf, 2014).

When studying the validation cohort, we verified that miR-3613-3p was in significantly low levels in urinary sediments of IgAN patients compared to healthy controls and MN or MCD patients, indicating it may have potential diagnostic value for IgAN. But it is necessary to note that the pathological grades of IgAN patients enrolled in this study were predominantly grade II (16.7%), III (51.0%), and IV (23.5%) according to Lee’s grading system, while grades I (3.9%) and V (4.9%) were rare. This phenomenon is also consistent with previous studies (Bartosik et al., 2001; Lee et al., 2005). In the present study, we also observed that the expression levels of miR-3613-3p and miR-4668-5p were significantly correlated with eGFR, Lee’s grades and glomerulosclerosis, indicating that there may be a possible correlation between the levels of the two miRNAs and the severity of IgAN.

New evidences are emerging that IgAN is an immune-mediated disease with galactose-deficient IgA1, which can elicit an autoantibody response and formation of immune complexes that are deposited in the mesangium (Kiryluk & Novak, 2014; Pillai, Balabhadraputani & Bhat, 2014). Although miR-3613-3p and miR-4668-5p were reported with unknown biological functions, bioinformatics analysis may shed light on the roles of these miRNAs by predicting miRNA-regulated genes. According to the GO analysis, the predicted functions of both miR-3613-3p and miR-4668-5p were significantly enriched in regulation of transcription and macromolecule metabolic process, which is consistent with the study of Tan et al. (2013). Moreover, the gene functions predicted by miR-3613-3p was also significantly enriched in B cell differentiation and activation, which may be related to the mechanism of IgAN.

According to the KEGG analysis, the predicted pathways of both miR-3613-3p and miR-4668-5p were significantly enriched in Wnt signaling pathway. It was reported previously that Wnt proteins are mitogenic for pro-B cells (Reya et al., 2000). Moreover, a recent study has shown that the Wnt pathway was hyper-activated in PBMCs of IgAN patients, and the proliferation rate of PBMCs isolated from IgAN patients was significantly enhanced (Cox et al., 2010). Therefore, we make a hypothesis that the decreased expression of miR-3613-3p might regulate the activation of B cells through Wnt pathway, and participate in the pathogenesis of IgAN, providing a direction for further functional studies.

There are a few limitations of our study. First, we detected urinary levels of candidate miRNAs without determining their cellular source. MiRNAs in urinary sediments presumably reflect cellular miRNA content, including podocytes, inflammatory cells, deciduous renal tubular epithelial cells and cells from urinary tract, but not including exosomes because of the relatively low g values during centrifugation. Future studies would be necessary to investigate cellular sources of candidate miRNAs. Secondly, as the bioinformatics analysis is notoriously inaccurate, the underlying mechanism of the changes and correlations observed in this study needs further investigation.

In summary, the expression profile of miRNAs was significantly altered in urinary sediments from IgAN patients, and the microarray data provided a valuable repertoire to discover non-invasive biomarkers for IgAN. Further studies with larger numbers of patients and controls, especially in different races, are urgently needed to validate the expression profiles. Urinary expression of miR-3613-3p was down-regulated in patients with IgAN, and the levels of both miR-3613-3p and miR-4668-5p were correlated with disease severity. Further studies are needed to explore the roles of miR-3613-3p and miR-4668-5p in the pathogenesis and progression of IgA nephropathy.

Supplemental Information

Figure S1 Volcano plots of miRNAs expression determined by miRNA microarray analysis

Urinary miRNAs levels were analyzed by miRNA microarray in the screening cohort, which including 6 patients with IgAN grade I–II, 6 patients with IgAN grade III, 6 patients with IgAN grade IV–V, 4 patients with MN, 4 patients with MCD and 6 healthy controls. Red and green dots represent the number of miRNAs that were significantly up-regulated and down-regulated, respectively, and black dots represent a lack of differential expression. Thethreshold of statistically significant difference was set at P < 0.05 and FC > 2. IgAN, IgA nephropathy; I–II, III and IV–V, grade I–II, III and IV–V according to Lee’s grading system, respectivley; HC, healthy control; MN, membranous nephropathy; MCD, minimal change disease; FC, fold change.

Click here for additional data file.

Table S1 Differentially expressed miRNAs in IgAN compared against HC

IgAN, IgA nephropathy; HC, healthy control; FC, fold change.

Click here for additional data file.

Table S2 Differentially expressed miRNAs in IgAN compared against MN

IgAN, IgA nephropathy; MN, membranous nephropathy; FC, fold change.

Click here for additional data file.

Table S3 Differentially expressed miRNAs in IgAN compared against MCD

IgAN, IgA nephropathy; MCD, minimal change disease; FC, fold change.

Click here for additional data file.

Table S4 The significant GO Biological Processes for miR-3613-3p

GO, Gene Ontology; BP, biology process.

Click here for additional data file.

Table S5 The significant GO Biological Processes for miR-4668-5p

GO, Gene Ontology; BP, biology process.

Click here for additional data file.

Supplemental Information 1 Additional information regarding the qPCR analysis procedure 6

Click here for additional data file.

Supplemental Information 2 Proof of consent form

Click here for additional data file.

Additional Information and Declarations

Competing Interests

Author Contributions

Human Ethics

Microarray Data Deposition

The authors declare there are no competing interests.

Nannan Wang conceived and designed the experiments, performed the experiments, analyzed the data, contributed reagents/materials/analysis tools, wrote the paper, prepared figures and/or tables, reviewed drafts of the paper.

Ru Bu performed the experiments, analyzed the data, reviewed drafts of the paper.

Zhiyu Duan performed the experiments, reviewed drafts of the paper.

Xueguang Zhang and Pu Chen performed the experiments, contributed reagents/materials/analysis tools, reviewed drafts of the paper.

Zuoxiang Li and Jie Wu contributed reagents/materials/analysis tools, reviewed drafts of the paper.

Guangyan Cai and Xiangmei Chen conceived and designed the experiments, reviewed drafts of the paper.

The following information was supplied relating to ethical approvals (i.e., approving body and any reference numbers):

1. the Ethics Committee of the Chinese PLA General Hospital.

2. S2014-004-02.

The following information was supplied regarding the deposition of microarray data:

GEO: GSE64306.

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
