# Peer review of "Profiling and initial validation of urinary microRNAs as biomarkers in IgA nephropathy"

_PeerJ, doi:10.7717/peerj.990_

## Round 0.1 · original submission · Major Revisions

This manuscript has been reviewed by two referees, one of them being the academic editor, whose comments are noted immediately below. The other reviewer's comments are noted further below. Please adequately address all comments in both reviews in the resubmission. The comment regarding corresponding authorship can be ignored. I strongly advise you to re-analyze the data with the IgAN cases graded as per the Oxford classification.

In this study, the authors have examined microRNAs in urine sediments to identify biomarkers associated with IgA nephropathy (IgAN). A microarray platform was used in the exploration phase of the study to identify microRNAs that are differentially expressed in IgAN grades I-II compared to healthy control. Two such microRNAs were subsequently validated in the second phase of the study, which used RT-PCR for microRNA profiling. The goal of the study and its use of a discovery and a validation phase are good. However, there are many issues with the work as presented in the manuscript. For example, it lacks information on the comparison of IgAN regardless of grade against control groups. Further, the division of IgAN into three sub-groups and then showing microRNA biomarker potential for the first but not the other two sub-groups, without adequate discussion in the manuscript text, seems arbitrary. The IgAN grading system that the authors use is outdated. In addition, the manuscript has not been prepared with appropriate thoroughness and clarity. E.g., the abstract is inadequately detailed, many figures and figure legends are unclear, and definitions of some of the groups of the study have not been properly provided. The various Results sections are written almost like summaries, without appropriate reasonings and descriptions.

1. Overall. A more up-to-date IgAN grading system should be used, and data re-analyzed.

2. Methods, 1st section. A statement on the ethics of the study (e.g., how was approval obtained) should be provided.

3. Methods, 1st section. How were 'healthy controls' and the 'MC' and 'MND' cases identified and chosen should be noted.

4. Methods, 2nd section. For microRNA RT-PCR assays, how much RNA was used in an RT reaction for subsequent PCR should be noted.

5. Results. Two-three sentences on the yield of RNA from urinary sediments should be provided. E.g., the average, range and SD of ng RNA per ml urine should be mentioned. Whether there was a difference between the different groups of the study (IgAN of different grades, HC, MN, etc.) should also be noted. Finally, a note on the nature of the urinary sediment should be provided. E.g., that they do not have exosomes (because of the relatively low g values during centrifugation).

6. Results, 2nd section. The section should also be re-written to explain what the 'screening cohort' is... how many cases it had... provide a summary of the microRNA profiling method... how many microRNAs were quantified.... how many were identified as differentially expressed and as per what criteria, and so on.

7. Results. Unsupervised and not just supervised hierarchical clustering should also be done, and the heat-map of microRNA expression shown for this analysis.

8. Results. What was the result of comparing all IgAN against controls, in both the exploration and the validation parts of the study? Presumably, no microRNA is differentially expressed in all IgAN grades are considered as one group and compared against non-IgAN groups. This should be clearly mentioned and discussed in detail, as well as noted in the Abstract.

9. Results, last section. Bioinformatic analysis of pathways/processes should be provided for both miRNAs, miR-223 and -3613, even if it requires that for -3613, three instead of six target prediction algorithms are required to return a candidate target mRNA. Additionally, the result section should provide more details. What were the numbers of target mRNAs identified by each algorithm? What were the criteria to identify the top 20 pathways/processes? Why only top 20? How many pathways/processes were identified?

10. The Abstract must note the number of samples/cases of each group for both screening and validation phases of the study. It must also note effect sizes. E.g., "miR-223-3p was significantly higher" should be qualified with the fold-change value. That 'I-II" in "IgAN I-II", for example, means grade should be mentioned along with the grading system. The Abstract also needs to be re-written for clarification. E.g., what does "miRNA-target database may provide a novel understanding" mean? Unnecessary details that are not very pertinent or uncertain, such as that on bioinformatic analysis of miR-223 targets (target prediction algorithms being notoriously inaccurate), should be removed. Did the two microRNAs differ among the early, mid and late IgAN grades should be noted in the Abstract.

11. Please italicize all microRNA names (and the name of any RNA or gene, but not protein). This should be done in both text and figures.

12. All abbreviations in Abstract should be explained (expanded on first use within the Abstract). E.g., "SAM" and "RT-qPCR".

13. First letters of some words are inappropriately upper- or lower-cased throughout the manuscript. This needs to be corrected.

14. Please use 'phase' instead of 'cohort' to refer to the exploration/discovery and validation parts of the study.

15. Examples include: "the Screening Cohort" in Table 1 (correct use should be "Screening cohort"); "Volcano Plots" and "Venn Diagram" in the 2nd Results section (should be "volcano plot" and "Venn diagram"); "Since Grades I-II" in first sentence of second paragraph of the 2nd Results section (should be "Since grades I-II); "MEAN±SEM" in Table 1 (should be "mean ± SEM"); and, "Whisker-box plot" in Figure 4 legend (should be "whisker box plot").

16. Results, 2nd section. The data deposition information should be moved to Methods.

17. The "explanation" for Table 1 should be provided in a footnote instead of in the paragraph at the top. Besides data, tables should only have titles and footnotes.

18. Figure 1 should be redrawn so that 'FC' and 'P value' are used instead of 'Ration' and 'P-value' in the axis titles. In addition, the figure legend must explain the abbreviations such as MN and FC, and also provide the group sizes.

19. Figure 2. The legend should provide more details to clarify what "differentially expressed" means.

20. Figure 3. The complete meaning of the color scale should be provided. Names of microRNAs should be noted. What the suffixes in the sample names mean, such as "LD" in "HC_LD", should be mentioned.

21. Figure 4. Group sizes and meaning of abbreviations like NS and MN should be noted in the legend. What the box-plots depict (mean, quartile, etc.) should also be noted. The meaning of NS as being P > 0.05 should be noted. The Y-axis title should be simplified by using just "Relative miR-X" instead of "urinary miR-X expression...".

22. Figure 5. The figure should be redrawn, e.g., to remove "miR-223" text from the panel, change "lgP" to "Log(P value)", etc. The legend should note how the top 20 processes/pathways were identified (criteria? P value?).

23. Figure S1. This figure to note validation of microarray data by RT-PCR needs to be completely redrawn. The figure, its legend, usage of terms like 'chip,' etc., are all inappropriate and unclear.

24. Figure S2. The figure and legend should be corrected as suggested for Figure 4.

·

Basic reporting

1. Only one corresponding author needs to be mentioned.
2. Pl provide IRB approval number if available.
3. For the RT-PCR please provide compliance with the MIQE guidelines and checklist

Experimental design

1. The research question is not clear - are the authors looking for a disease marker which distinguishes IgAN from normal healthy people or are they looking for a disease severity or disease outcome marker helpful for prognostication? The results seem to point towards the former. However this would be of limited relevance since IgAN is not defined by miRNA expression but on a constellation of clinicopahological findings.

2. The authors need to specify the selection methodology adopted for choosing the cases for various grades of IgAN. The number of biopsies during the period of study, eligible cases for inclusion and the final method for choosing the 18 IgAN cases must be clearly brought out.
3. Please specify how healthy controls were defined.
4. Catalog no of the various kits used must be annotated.
5. Please provide eGFR for the 2 cohorts of patients - it has a greter relevance in terms of outcomes.

Validity of the findings

1. The research question is not clear - are the authors looking for a disease marker which distinguishes IgAN from normal healthy people or are they looking for a disease severity or disease outcome marker helpful for prognostication? The results seem to point towards the former since the expression profiles of the final 2 miRNAs is no different between IgAN Grade I-II vs IgAN Gr IV-V. However this would be of limited relevance since IgAN is not defined by miRNA expression but on a constellation of clinicopahological findings.

2. The histologic classification of Lee et al has limited relevance to outcomes in clinical practice. The current Oxford classification is supposed to be more robust and clinically relevant. Why did the authors choose Lee's classification. Is it possible to look at the data from Oxford Classification and the MEST criteria?

3. The GO pathway results for mi-223 needs further discussion especially in view of lin with cancer pathways.

Additional comments

NIL

---

## Round 0.2 · Minor Revisions

The revised manuscript adequately addresses the concerns raised in the last round of review. However, it needs to be corrected to address the following issues:

1. There are many typographical errors. Such as 'heumatoid arthritis,' and 'ration' instead of 'ratio', and sentence beginning with 'miRNAs' instead of 'MiRNAs.' Please look for and correct such errors.
2. Abstract: In 'differently expressed miRNAs in IgAN', change 'differently' to 'differentially.'
3. Abstract: For the correlation coefficient values, please add the an indicator of the type of test. E.g., '(Pearson r = XX, P < YY).' This may be done only for the first instance of 'r' being mentioned if all correlation analyses used the same test.
4. Abstract: Similarly, as above, for the P values of inter-group comparison, note the type of test.
5. Abstract: In conclusions, 'patient' should be plural 'patients.'
6. Figure 5 and 6 legends: Please add a note on what 'lg' refers to (log2? log10?).
7. Table 4: Italicize microRNA names and remove the 'hsa' prefix.
8. Table 5: Italicize microRNA names and add '3p' or '5p' suffixes.

---

## Round 0.3 · accepted · Accept

Thank you for submitting the revised version of the manuscript to address the largely language/presentation-related errors that I had pointed to. The manuscript is now acceptable for publication.